# Contact Resistance Sensing for Touch and Squeeze Interactions

**Nianmei Zhou** [1,*] **, Steven Devleminck** [1,2] **and Luc Geurts** [1,*]

1  KU Leuven, 3000 Leuven, Belgium; steven.devleminck@kuleuven.be
2  LUCA School of Arts, 1030 Brussels, Belgium
*  Correspondence: nianmei.zhou@kuleuven.be (N.Z.); luc.geurts@kuleuven.be (L.G.)

**Abstract:** This study investigates accessible and sensitive electrode solutions for detecting touches and squeezes on soft interfaces based on commercially available conductive polyurethane foam. Various electrode materials and configurations are explored, and for electrodes made of conductive threads, the static and dynamic electrical behaviors are studied in depth. In contrast to existing approaches that aim to minimize or stabilize contact resistance, we propose leveraging contact resistance to significantly enhance sensing sensitivity. Suggestions for future researchers and developers when building squeeze sensors based on this material are provided. Our findings offer insights for DIY enthusiasts and researchers, enabling them to develop sensitive soft interfaces for touch and squeeze interactions in an affordable and accessible manner and provide a completely soft user experience.

**Keywords:** soft sensor; tangible user interface; deformable interface; conductive foam; DIY; squeeze interaction; squeezable interface

## 1. Introduction

The majority of tangible interfaces currently in use are rigid, whereas soft interfaces present opportunities for expressive and playful interactions. Soft interfaces are regarded as more natural and expressive [1]. Researchers have explored the potential of soft interfaces in various domains, such as music instruments [2–6], breathing sensors [7], modeling tools [8–11], toys [12,13], controllers [11–17], mental health [18–20], communication [21,22], furniture [13,23], and robot skin [24]. These interfaces come in different sizes, ranging from finger interaction [14,15,25] to hand interaction [16,24,26] to whole body interactions [13,27]. With regard to sensing mechanisms, several novel sensing materials or new strategies for building soft sensors have been proposed [12,28–31]. However, these approaches are often relatively complex to implement and might require access to chemistry labs and costly materials. Access to a chemistry laboratory is not always necessary. Some systems utilized commercially available materials to build squeeze sensors. Some of them are embedded with rigid sensors to detect compression [2,5,13,14,17,19,21,23,24,26,32–35]. By contrast, to remove the rigid components from the compression part of the sensor, some studies utilized commercially available soft conductive materials to detect the deformation of the soft interface. Conductive polyurethane (PU) foam [8,9,15,27,36] and conductive wool [6,16,27,37,38] are two typical materials that have been used for this purpose.

In sensing solutions based on conductive foam, the contact resistance between electrodes and foam plays a significant role, due to the porous structure of foams [39]. The total resistance of such a sensor is the sum of the resistance of the electrodes, the contact resistance between electrodes and the material, and the resistance of the foam material itself [40,41]. One common approach is to reduce or stabilize contact resistance by building strong mechanical connections between electrodes and the foam and/or by increasing the contact area [7,12,39]. Alternatively, the contact resistance can be leveraged to improve the sensitivity of foam-based strain/pressure sensors [40–43], but these electrode solutions need a background in material science and access to a chemistry laboratory, and thus

are more difficult to be adopted by Human–Computer Interaction (HCI) researchers and do-it-yourself (DIY) lovers.

The main goal of this paper is to investigate the static and dynamic resistance behaviors of accessible electrode solutions for soft interfaces based on commercially available conductive PU foams. Instead of high-tech solutions that require expensive materials or access to a chemistry lab, our goal is to provide and assess simple and cost-effective approaches to constructing soft interfaces. Three main experiments were carried out to explore the electrical properties of sensors based on conductive foam. The main objectives were as follows:

- Pilot experiment: to examine the contribution of contact resistance on the total resistance change during compression.
- Experiment 1: to investigate the static resistance properties of samples using six electrode solutions based on three accessible materials—copper tape, conductive fabric tape, and conductive threads.
- Experiment 2: to investigate the dynamic resistance behavior of foam sensors based on two electrode solutions utilizing conductive threads.

In the following chapters, related works will first be discussed. Then, an overview of the experimental setup will be introduced. After that, descriptions and discussions of our experiments will be presented. Subsequently, two applications using our sensing solutions will be introduced. Then, we will discuss the implications of our experiments and suggestions for future researchers and DIY enthusiasts when building soft sensors and processing signals. Finally, the conclusion and limitations of our work, and potential future research directions are outlined.

## 2. Related Work

### 2.1. Sensing for Squeezable Interfaces

There are a number of studies on squeezable interfaces in which rigid sensors are embedded. Some typical sensing technologies are optical sensing [13,14,21,24,34,35], air pressure sensing [44,45], force sensing [2,5,17,19,32,33] and acoustic sensing [23,26]. However, if a completely soft experience is expected, hard components could distract from the experience of softness when squeezing. Other researchers have proposed soft conductive materials as strain/pressure sensors, such as conductive wool [6,16,27,37,38] and conductive foam [8,9,15,27,28,39,46]. One interesting characteristic of these material is that their resistance decreases when being pressed. Conductive threads [47–49] were also used in some studies as pressure sensors.

Most of these foam-based sensors are based on the piezoresistive sensing mechanism [41,50,51], while capacitive sensing is also possible [46,52]. In the former case, when conductive foam is compressed, the conductivity of the foam material increases because the contact areas within the porous structure of conductive foam increase, thereby producing more conductive pathways [41,50,51]. The total resistance consists of three components: (1) the resistance of the electrodes, (2) the contact resistance between the electrodes and the foam, and (3) the resistance of the foam [40,41]. When a force is applied, the load exerted on the sensor and increased contact area between the electrodes and the conductive material lead to a decrease in contact resistance [41,53].

There have been a number of studies focusing on the fabrication and evaluation of novel conductive foam materials [12,28,39,52,54–56]. However, these approaches are less suitable for developers who lack access to chemical laboratories or who prefer a faster and simpler method of prototyping by utilizing off-the-shelf products. As a result, commercially available conductive PU foam, a sub-category of ESD (electrostatic discharge) foam, which is mainly used as a packaging material to protect electrostatic-sensitive devices, has been used in several studies to develop deformable interfaces [8,9,15,27]. This material is more accessible and can be easily purchased online. Electrode solutions play a vital role in the resistance behavior of foam-based sensors [39], and different electrodes can result in completely different static and dynamic resistance behavior [39]. Despite the wide

use of conductive foam in the HCI community, not much attention has been paid to the electrode configuration.

### 2.2. Electrodes for Foam Sensors

Some HCI researchers used accessible soft conductive materials as electrodes, such as conductive thread and conductive fabric. FoamSense [12] wrapped the conductive threads around the foam. Laying a sheet of conductive fabric on the top and/or the bottom of a piece of conductive foam is also a commonly used method [7,9,15,27,28,56]. In some studies, copper wires were directly put into the conductive foam as electrodes [4,7]. Wang et al. [39] used stretchable silver paste to reduce the contact resistance between the conductor and the electrodes. Nakamaru and colleagues [12] also discussed solutions to decrease contact resistance for the foam sensors they proposed. So, most of the proposed solutions use relatively stable mechanical connections or larger contact areas, thus reducing and/or stabilizing the contact resistance. The previous literature suggests that contact resistance can enhance electrode sensitivity [40–43]. This study aims to explore the potential of utilizing contact resistance as the main sensing mechanism and characterizing the properties of contact-resistance-based electrode solutions.

## 3. Test Setup

A data acquisition setup based on Arduino UNO was used to measure resistance values (Figure 1a), and a compression setup controlled the deformation of the samples (Figure 1c).

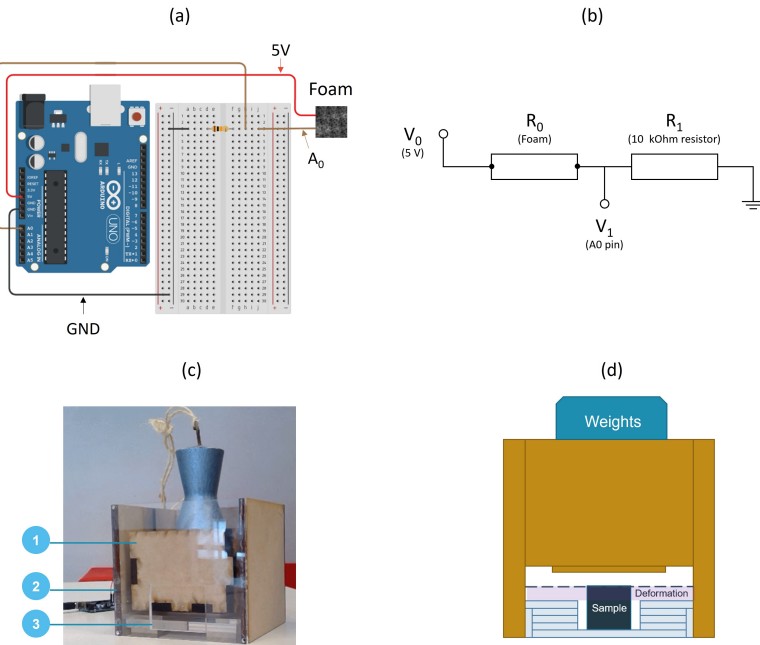

**Figure 1.** Experimental setups: (**a**) schematic of the data acquisition setup based on Arduino; (**b**) rationale of data acquisition; (**c**) compression setup: 1—a box with weight, 2—a larger wooden box with tracks, enabling part 1 to fall vertically, 3—plexiglass plates used to determine deformation values; (**d**) rationale of the compression setup.

### 3.1. Data Acquisition Setup

To investigate the change in the resistance of conductive PU foam samples over time, an Arduino UNO, together with a breadboard and a resistor (10,000 ohms), were used to measure the resistance of the foam samples (Figure 1a). A resistor of 10,000 ohms was used because, in preliminary measurements of the foam cubes using a multimeter, the optimal value of the resistor $R_1$ in Figure 1b was calculated according to the formula: $R = \sqrt{R_{max} \times R_{min}}$, where $R_{min}$ is the lowest resistance when the sample is compressed, and $R_{max}$ is the largest resistance during compression. Such a formula allows us to maxi-

mize the input voltage range. A voltage divider was used to obtain the resistance values of the foam sample ($R_0$ in Figure 1b), whose resistance decreases when deformed.

*3.2. Compression Setup*

As shown in Figure 1c, a customized instrument was used to control the strains and loads. It consists of three components.

- The primary function of the first component is to provide compression and load control during the deformation process (Figure 1c, 1). Weights are put in a box to exert compression in most experiments. To facilitate vertical descent, the wooden box is accompanied by two elongated bars on each side, designed to align with the tracks of the big wooden box. A box made of paper is used to apply small loads in Experiment 1 (i.e., 0.06 N and 1.04 N) to minimize the weight of the container itself and provide an even contact surface.
- The second component is a larger wooden box (Figure 1c, 2) which is strategically designed to facilitate the controlled vertical descent of the first component (the box). This larger wooden apparatus is equipped with two parallel tracks, situated on both the left and right sides.
- The third component serves to establish and control the desired deformation. It consists of plexiglass plates with a thickness of 5 mm (Figure 1c, 3). By adjusting the height and order of the plates, the foam sample's deformation can be controlled (Figure 1d).

The compression setup is not connected to the Arduino. When Component 2 is dropped, compression is exerted on the foam sample. As illustrated in Figure 1d, the load is controlled by the weight in Component 2, and the strain is controlled by Component 3 according to the height and order of the plexiglass plates.

## 4. Pilot Experiment: Conductive Wool vs. Conductive PU Foam

This pilot test aims to compare the durability of commercially available conductive wool and conductive PU foam by measuring the shrink of sample length after multiple rounds of compression. A sample of 24 grams of wool (Bekinox, W12/18) and a sample of PU foam of 70 mm height were used and compressed by 40% of their sizes 150 times, and the change in length of the sample was measured.

After 150 rounds of compression, the length of the wool sample was reduced by 28%, whereas the length of the foam sample was reduced by 6% (Figure 2). The decrease in size in conductive wool is likely the result of intensive use, which causes the fibers to begin to stick with each other. By contrast, conductive polyurethane (PU) foam is more durable in terms of shape-keeping ability after intensive compression.

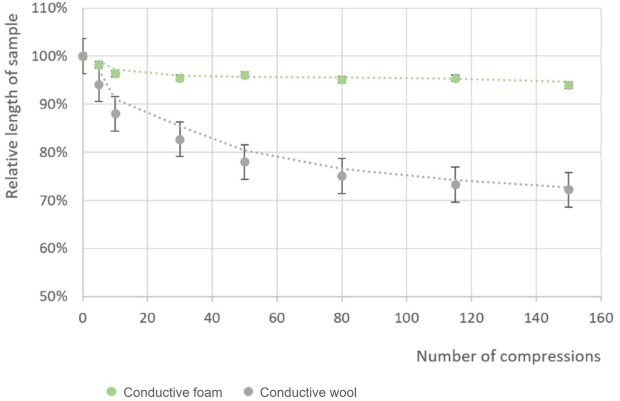

**Figure 2.** Relative length change under compression. The green line is the relative length of *conductive foam* samples after x rounds of compression. The gray line is the relative length of *conductive wool* samples after x rounds of compression.

### 5. Pilot Experiment: Contact Resistance

The goal of this experiment is to verify which component is the primary cause of the change in resistance when the conductive PU foam sensor is deformed—the contact resistance or the resistance of the conductive foam itself because of volume change. Two hypotheses were proposed:

- Hypothesis 1: The major cause of resistance change is the change in the volume of the foam.
- Hypothesis 2: The major cause of resistance change is the contact between the electrodes and the foam.

Two samples of conductive PU foam (Desco Industries Inc., 241520) were used (Figure 3a) to test our hypotheses, whose sizes were around 39 cm (length) × 8 cm (width) × 3 cm (height). For each sample, two conductive threads were inserted at each end of the foam as electrodes (total length: 18 cm, inserted length: 3 cm). A pile of plexiglass with a total height of 15 mm was placed on each side of the foam to achieve a deformation of 15 mm and make sure the contact areas between the foam and the weight were identical. The weight of each pile of glasses was 77.1 grams. A 5 kg weight was placed at three different locations: directly on an electrode (p1 in Figure 3b) between the center of the foam and p1 (p2 in Figure 3b), and at the center (p3 in Figure 3b). Our assumptions were:

- If the changes in resistance were similar in all three circumstances, then the change in volume of the foam would be the major factor causing the change in resistance, therefore suggesting H1;
- If the change in resistance was more significant as the weight was placed on the electrode, then the main factor influencing the resistance would be the contact resistance, therefore suggesting H2.

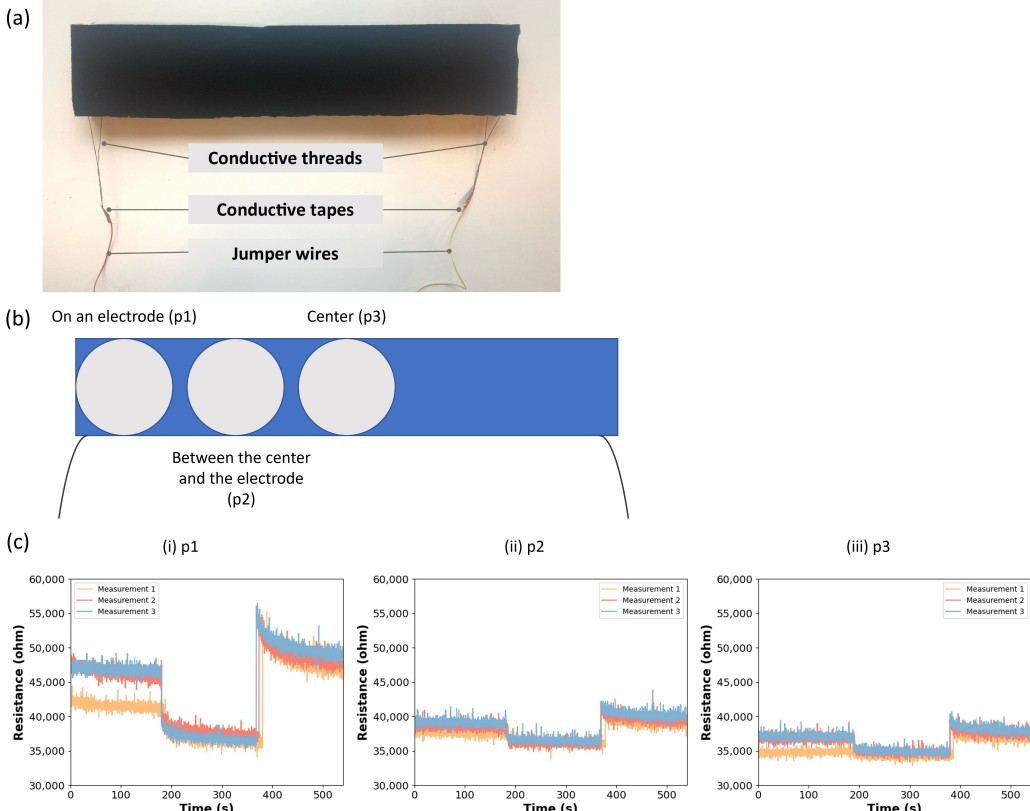

**Figure 3.** Contact resistance test: (**a**) components of the foam sample; (**b**) three locations of compression; (**c**) resistance over time when a 5 kg weight was put on p1, p2, and p3.

*5.1. Procedure*

The resistance of the foam was first measured when it was not deformed (duration: three minutes), then manually placed the weight on one of the three locations shown in Figure 3b for three minutes to observe the sample's resistance behavior during rest and compression. Finally, the weight was removed, and the resistance after deformation was recorded for three minutes. A three-minute duration was used for each stage to observe the resistance change during a relatively long duration. The above steps were repeated three times for each location. The resistance data were plotted on a timeline.

*5.2. Results*

The change in resistance was most pronounced when the weight was placed on p1 (electrode) (Figure 3c, i), while less significant when the compression was at p2 (between the electrode and p3) (Figure 3c, ii) and p3 (center of the foam) (Figure 3c, iii).

Moreover, several dynamic features were observed in the resistance change: a continuous decrease in resistance after the onset of compression, a surge at the end of the compression, and a slow recovery after compression. After the surge, the resistance values slowly decreased and settled to a value that was higher than its original value before the compression. A slight increase at the onset of compression was also sometimes observed (Figures 3c and A1). Another noticeable fact is that their resistance values during compression were slightly different, though the measurements were taken from the same sample.

For clarity and conciseness, data from the other sample are included in Appendix A.

*5.3. Discussion*

Since the resistance changed most significantly when the weight was put directly on the electrode, we argue that our results support H2. This means that the contact resistance between the electrode and the foam plays a more important role than the resistance change in the foam itself.

Drawing from this finding, there are two different directions to explore electrode solutions: (a) decreasing or stabilizing contact resistance by increasing the contact area between the foam and the electrodes and/or creating strong mechanical connections between the foam and the electrodes; (b) exploiting contact resistance to increase sensitivity.

Regarding its dynamic resistance behavior, the initial rise in resistance at the onset of compression can be due to the initial compression breaking some of the conductive pathways inside the foam and the contact area between electrodes and the foam. The same factor can cause a surge in resistance after compression. The continuous decrease in resistance value after the onset of compression should be caused by the creep behavior of foam. Such behavior was also reported in other studies involving foam-based sensors [57,58], which means the size of a piece of PU foam tends to keep decreasing under constant pressure. The drifted baseline and slow recovery were observed in other foam-based pressure sensors [57,59], together with the difference in resistance in different rounds of measurements [39,59]. These behaviors can be attributed to the hysteresis effect of foam [59]. This effect is that, after being compressed, it tends to take a long time for this material to return to its original baseline due to the loss of the internal energy [59]. Hence, the resistance recovers slowly and different rounds of measurements have different baseline resistance values.

## 6. Experiment 1: Electrode Solutions

This experiment compared six affordable electrode solutions using off-the-shelf materials. The electrode solutions are based on three off-the-shelf materials: copper tape, conductive fabric tape, and conductive threads. Their performance under different strains and loads was tested. Sensitivity, stability, and repeatability are the main criteria for the evaluation of these electrode solutions. In addition, the tactile sensation of the electrodes

and the supported interaction types (types of interaction that can be performed in such samples and are detectable) are also part of the evaluation criteria.

### 6.1. Sample Preparation

Similar to in the pilot experiment, a carbon-impregnated open-cell flexible polyurethane foam (Desco Industries Inc., Chino, CA, USA, 241520) was used as the core material. The foam sheet was cut into cubes of 30 mm × 30 mm × 30 mm by a sawing machine (SENAS Inc., Pavia, Italy, 400). Regarding the electrode solutions, affordable materials and simple fabrication methods were intentionally chosen that do not require access to a laboratory environment. Conductive adhesives were excluded from consideration due to their relatively high prices (e.g., MG Chemicals 8330S: EUR 06.61/21 g; DuPont™ PE873: EUR 455/100 g). Six accessible electrode solutions were tested (Figure 4) whose prices are under 0.014 euros per centimeter. The configurations of the materials used as a single electrode are as follows:

(a) Conductive copper tape (AT528, Advance Tapes, Leicester, UK): 19 mm × 30 mm;
(b) Conductive fabric tape (MDFT-10F-1I, MOS Equipment, Santa Barbara, CA, USA): 19 mm × 30 mm;
(c) One piece of conductive thread inserted into the surface of foam (641, Adafruit Inc., New York, NY, USA): 160 mm;
(d) Two pieces of conductive threads inserted into the surfaces of the foam (641, Adafruit Inc., New York, NY, USA): 160 mm × 2 pieces. The distance between two conductive threads that constitute the same electrode is 10 mm;
(e) One piece of conductive thread loosely put on the surface of the foam (641, Adafruit Inc., New York, NY, USA): 160 mm;
(f) Two pieces of conductive thread loosely put on the surface of the foam (641, Adafruit Inc., New York, NY, USA): 160 mm. The distance between two conductive threads that constitute the same electrode is 10 mm.

Identical electrodes were put on the top and bottom surfaces of each sample. Copper tape (solution a) is the most accessible option since it can be easily purchased at hardware or electronic stores. Solutions that utilize copper or fabric tapes aim to create a larger contact area between electrodes and the foam. Solutions (c) and (d), namely inserting conductive threads into the foam as electrodes, aim to establish a more stable mechanical connection between the electrodes and the foam. By contrast, by placing conductive threads loosely on the surfaces of the foam, solutions (e) and (f) attempt to leverage the contact resistance to enhance the sensitivity of the sensor to light touches and soft squeezes. Since the contact resistance is greatly affected by the contact area, these two solutions magnify the change in contact area by loosely placing conductive threads on the top and bottom surfaces of the foam. In solutions (c)–(d) and (e)–(f), the objective was to investigate the impact brought by the number of threads inserted as electrodes, since theoretically, inserting more threads will result in a larger contact area. Note that the two pieces are connected outside the samples, creating two resistors in parallel.

For each electrode solution, three identical samples were made to test the consistency and avoid the possible biases caused by only one sample. To fabricate an electrode using conductive tapes (solution (a) and (b)), one end of a copper wire was stripped (stripped length: 35 mm) and put underneath the conductive tape, and connected the other end to the circuit. For electrodes (c) and (d), the conductive threads were inserted closely along the surface of the foam; the thickness of the thread was 0.25 mm, with an insertion length of 30 mm. The conductive threads were knotted in one end to avoid displacement of the electrodes. For the same reason, to fabricate electrodes (e) and (f), a non-conductive thread was inserted into the foam surface, tied on both ends to prevent conductive threads from detaching from the foam surface, and the knots stuck out of the foam (2 mm) to produce a loose contact. A layer of isolated tape was wrapped around the conductive threads in electrodes (c), (d), (e), and (f) that were left on the outside of the foam to avoid short circuits.

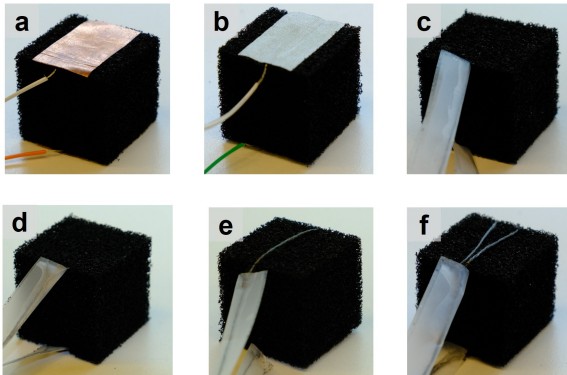

**Figure 4.** Electrode solutions: (**a**) conductive copper tape; (**b**) conductive fabric tape; (**c**) one piece of conductive thread inserted into the foam; (**d**) two pieces of conductive thread inserted into the foam; (**e**) one piece of conductive thread placed on the surface of the foam; (**f**) two pieces of conductive thread placed on the surface of the foam.

### 6.2. Procedure

In this experiment, pressure was applied in such a way that the distance between the electrodes was reduced. The experiment consisted of two parts: a strain test (Test 1) and a load test (Test 2). The load and strain were controlled by the setup described in Section 3.2.

In Test 1, a weight of 6 kg put into a box of 0.367 kg was utilized to create deformation. First, the resistance of the sample was recorded for one minute without compression. Next, pressure was applied by releasing the weight and the resistance values were recorded during a one-minute interval. After that, the load was removed and the resistance was recorded again for one minute. The sampling rate on Arduino was 100 Hz. For each sample and each strain value, this process was repeated three times to allow us to assess the repeatability of measurements within the same sample. The strain values are calculated as follows: $\Delta L/L_0 = (L_0 - L)/L_0 \times 100\%$, in which $L_0$ is the initial length of the foam and L is the length of the foam under deformation [60]. Six strain values were used in this test: 1%, 17%, 33%, 50%, 69%, and 83%. This resulted in 6 (strain values) × 3 (samples) × 3 (repetitions) = 54 measurements for each electrode solution in this test.

In Test 2, the load F in Newton is calculated by F = mg, where m is the total weight of the box and the weights (kg), and g is the gravitational constant 9.8 m/s$^2$. Similar to Test 1, the resistance values before, during, and after deformation were recorded, with each phase lasting one minute. Six load values were used in this test: 0.06 N, 1.04 N, 3.06 N, 13.06 N, 23.06 N, and 63.06 N. This resulted in 6 (load values) × 3 (samples) × 3 (repetitions) = 54 measurements for each electrode solution in this test.

### 6.3. Methods of Analysis

Our analysis mainly focused on the static properties of different electrode materials. As stated previously, the sensitivity, stability, and repeatability of each sensor solution are assessed.

Raw data from Tests 1 and 2 were first segmented in Python to obtain the resistance data before, during, and after compression. The data before and during compression were segmented based on the data point when the compression started, extracting the data from 30 s before and after that date point.

The average resistance of samples was first calculated when they were not deformed for each electrode solution over the 108 measurements in both tests, denoted as $R_0$. No filtering was no filtering procedures were applied to the acquired signals.

To assess the **sensitivity** of different electrode solutions, the ratio of resistance change for each measurement was computed, according to the following formula [60]:

$$\frac{\Delta R}{R_0} = \left(\frac{R_1 - R_0}{R_0}\right) \times 100\% \tag{1}$$

in which $R_0$ is the average resistance before compression and $R_1$ is the average resistance during compression. The first 1500 data points were chosen to calculate $R_0$, which corresponded to the first 15 s of the segmented data before compression, because they are not influenced by the slight increase in resistance at the onset of compression (Figure 3c). The last 1500 data points were chosen during compression to calculate $R_1$, because they are unaffected by the initial continuous decrease in resistance during compression (Figure 3c), and thus are relatively more stable.

To compare the **stability** of different electrode solutions, the following values are calculated:

- The coefficient of variation (CV) of the first 1500 data points before deformation, defined by the ratio of standard deviation (SD) to the mean value, denoted as $CV_{R_0}$.
- The CV of the last 1500 data points during compression, denoted as $CV_{R_1}$.

To assess the **repeatability** of the measurements within the same sample, the following values are calculated:

- The SD of the $\Delta R / R_0$ values for each sample under the same strain in Test 1, denoted as $\sigma_{(}\Delta R_s / R_0)$.
- The SD of the $\Delta R / R_0$ values for each sample under the same load in Test 2, denoted as $\sigma(\Delta R_l / R_0)$.

### 6.4. Results

Samples with conductive threads inserted as electrodes (solutions c and d) exhibited the lowest resistance values when no compression occurred (Table 1). In contrast, samples with one or two pieces of conductive threads placed on the surface (solutions e and f) showed the highest resistance values in the same condition. This is not surprising, given the firm mechanical connection in the former case, and the loose connection between electrodes and foam in the latter case.

**Table 1.** $R_0$: the average value of resistance before compression; $CV(R_0)$: the average value of the coefficient of variance of resistances before compression; $CV(R_1)$: the average value of the coefficient of variance of resistances during compression; $\sigma(\Delta R_s / R_0)$: the average SD of the resistance change ratios within the same sample under the same strain in Test 1; $\sigma(\Delta R_l / R_0)$: the average SD of the resistance change ratios within the same sample under the same load in Test 2. The lowest value of each column is represented in bold and italic.

| Electrode | $R_0$ (ohm) | $CV(R_0)$ | $CV(R_1)$ | $\sigma(\Delta R_s/R_0)$ (%) | $\sigma(\Delta R_l/R_0)$ (%) |
|---|---|---|---|---|---|
| a. Copper tape | 18,801 | 0.012 | 0.012 | 4.19 | 3.09 |
| b. Fabric tape | 12,418 | 0.011 | 0.013 | 4.26 | 2.05 |
| c. Thread (insert, 1 piece) | 11,293 | 0.005 | ***0.005*** | 3.09 | 2.75 |
| d. Thread (insert, 2 pieces) | ***11,120*** | ***0.004*** | 0.006 | 3.78 | 2.17 |
| e. Thread (surface, 1 piece) | 27,857 | 0.012 | 0.006 | 3.46 | 3.35 |
| f. Thread (surface, 2 pieces) | 85,735 | 0.017 | 0.007 | ***2.76*** | ***1.82*** |

**Sensitivity**. Figure 5 shows the resistance change ratios for different values of strain (Test 1) and load (Test 2).

The solutions placing conductive threads on the surface (e and f) exhibited a high sensitivity to small values of strain and load (Figure 5a,b). The solution that used two threads demonstrated a slightly better sensitivity than using one thread in most of the cases. Even at a minimal strain or load, there was an apparent change in resistance. Resistance decreased on average by 77% and 36% under the smallest strain of 1% and the smallest load of 0.06 N, respectively.

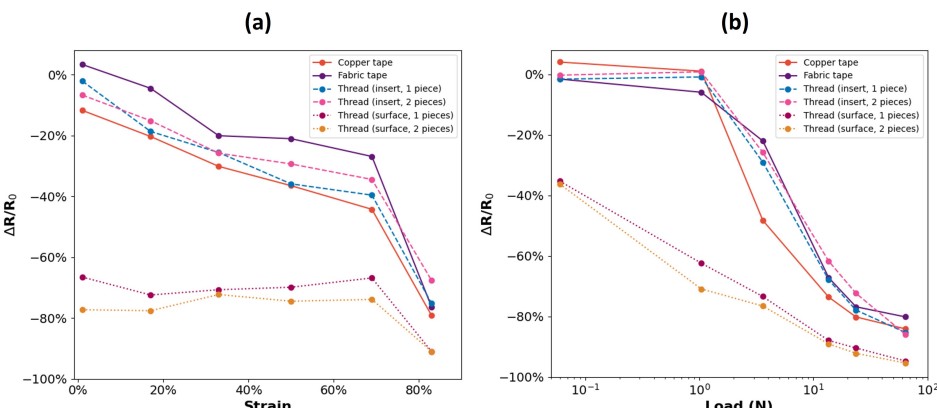

**Figure 5.** (**a**) Test 1: strain to $\Delta R/R_0$ curve; (**b**) Test 2: load to $\Delta R/R_0$ curve

In Test 1, the average $\Delta R/R_0$ exhibited a decreasing trend with the increase in strain in almost all solutions, except for solutions putting threads on the foam surface (Figure 5a). Nevertheless, in Test 2, the average values of $\Delta R/R_0$ of both solutions decreased with the increase in the load. Moreover, their $\Delta R/R_0$ exhibited the most linear relationship with log(N) compared to other solutions.

Other solutions had similar patterns of strain to $\Delta R/R_0$ curve and load to $\Delta R/R_0$ curve. Among them, the samples using fabric tape and inserted with two threads were relatively less sensitive compared to the other solutions (Figure 5a). In addition, the average resistance of samples using tapes even increased at small strains (fabric tape) or loads (copper tape).

**Stability**. As shown in Table 1, the lowest relative resistance fluctuation (CV) was observed when conductive threads were inserted as electrodes (solution c and d), in both cases before compression ($CV(R_0)$) and during compression ($CV(R_1)$). This can be attributed to the stable mechanical connection between electrodes and foam, which brought stable resistance data. By contrast, the samples using copper tapes and fabric tapes (solution a and b) exhibited high CV in both cases, with or without compression, and their CVs were over twice as high as those of the samples inserted with conductive threads. Interestingly, the solutions of placing threads on the surface of foam (solutions e and f) demonstrated high relative fluctuation $CV(R_0)$ when no compression happened. However, their average relative fluctuation of the resistance during compression was close to that of the solutions that inserted threads into the foam, namely solutions (c) and (d).

**Repeatability**. In general, this difference was not large among all solutions (the $\sigma(\Delta R_s/R_0)$ and $\sigma(\Delta R_l/R_0)$ in Table 1). Nevertheless, solution (f), namely placing two pieces of conductive threads on the surface, demonstrated the lowest average SD of $\Delta R/R_0$ for measurements taken from the same sample under the same strain or load. This implies that this electrode solution reacted more consistently to the same strain or load. By contrast, solutions (a) and (b), which were copper and fabric tape, respectively, exhibited relatively low repeatability under different strain values. In Test 2, solution (e), namely placing one piece of conductive thread on the surface as an electrode, had the largest change range of $\Delta R_l/R_0$ under the same load. Samples inserted with one or two pieces of conductive thread had a medium change range of $\Delta R/R_0$ in both Test 1 and Test 2. Additionally, the $\sigma(\Delta R/R_0)$ among measurements taken from different samples was relatively larger than that within the same sample (Table A1 in Appendix B).

*6.5. Discussion*

Based on the results of Experiment 1, our assessment of all electrode materials is summarized in Table 2.

**Table 2.** Assessment of each electrode solution. Tactile sensation means how it feels when samples with such electrodes are touched or compressed. Interaction type refers to the type of interaction that can be performed and is detectable in samples with such electrodes.

| Electrode | Cost | Sensitivity | Stability | Repeatability | Tactile Sensation | Interaction Type |
|---|---|---|---|---|---|---|
| Copper tape | 0.014 euro/cm$^2$ | Medium | Medium | Medium | Rigid | One dimensional |
| Fabric tape | 0.016 euro/cm$^2$ | Medium | Medium | Medium | Stiff fabric | One dimensional |
| Thread—insert | 0.007 euro/cm | Medium | High | Medium-high | Soft | Multi-dimensional |
| Thread—surface | | High | Medium | High | Soft | One dimensional |

The solutions using conductive threads were chosen for future experiments due to their satisfactory overall electrical properties and tactile features. The solutions involving the placement of conductive threads on the surface demonstrated the highest **sensitivity**, aligning with previous studies that utilized contact resistance to enhance the sensitivity of flexible pressure sensors [40,42,43]. These two solutions exhibited a significant resistance change even when the strain and load were very small (1% of strain and 0.06 N of load). This feature enables them not only to sense the hardness of squeezes but also to detect slight touches. Placing more conductive threads on the surface made the samples slightly more sensitive to the same strain or load (Figure 5). This can be because, when two conductive threads were placed on the surface as an electrode, there was a larger change in the contact area between the electrodes and the foam during compression. This solution demonstrates a high level of sensitivity compared to other solutions in our experiment and in a previous study [7]. Additionally, their strain to $\Delta R/R_0$ curves in Test 1 were relatively flat. This could be attributed to the fact that, in Test 1, the same weight was used in all experiments. Moreover, this solution had stable data when compression happened. Although placing more threads introduced more fluctuations when there was no compression, its **repeatability** of resistance change ratio in one sample was the best among all the solutions in both Test 1 and Test 2. However, its resistance change ratio varied a lot in measurements from different samples (see Appendix A). This should be attributed to different contact conditions between the foams and threads due to the fabrication process, which leads to the variation in their contact resistance and its change range. This suggests that calibration should be always carried out in different samples.

The samples using inserted threads had sufficient sensitivity to large strains over 17% and loads over 1.04 N. Moreover, they demonstrated high stability (i.e., low relative resistance fluctuation). This should be attributed to the stable contact area between the inserted threads and the foam, which produced less fluctuation of resistance data. The number of threads inserted into a foam did not significantly affect sensitivity. The samples inserted with two threads as an electrode were slightly less sensitive than those inserted with one thread in most of the strain or load values, even the former ones had a lower average resistance value when no compression happened. This might be caused by the fact that inserting two threads as an electrode created a larger contact area with the foam, thereby reducing the contact resistance, which is sensitive to force and the contact area between the electrode and the foam. However, the change in the resistance of the foam was not as significant as the change in the contact resistance under the same deformation or strain. As a result, the average change ratio of the resistance of samples inserting two threads as an electrode was lower than that of samples using one thread.

To our surprise, conductive fabric did not exhibit high sensitivity or stability. Theoretically, this solution should have a larger contact area with the foam, which should decrease the contact resistance between the foam and electrodes. The low stability may be attributed to the instability in the contact between fabrics and foam, despite the contact area appearing larger than the inserted threads; this, in turn, introduces unstable resistance values. In addition, the change in contact points between fabric and foam did not increase as dramatically as the solutions that placed threads on foam surfaces.

From the perspective of the tactile features, solutions using conductive threads were the only solutions that did not introduce any stiffness; thus, the samples using these solutions could be deformed in different directions without feeling the existence of the electrodes. This characteristic makes them suitable for the development of squeezable interfaces that can be touched and squeezed in all directions. Specifically, samples with inserted electrodes were able to sense compression in multiple directions, even when the distance between the electrodes was small (see Appendix C). By contrast, the samples placed with conductive threads on the surface were less sensitive to touches or squeezes that were not on the electrodes. In comparison, when squeezing a sample with conductive fabric, the slight stiffness of the conductive fabric can be felt, and so do copper tapes. In addition, copper and fabric tapes were wrinkled when the compression was not on the surface. As a result, the surfaces of the electrodes became uneven. Using conductive threads as electrodes could enable the foam to be deformed in any direction and was able to return to its original shape. Furthermore, it could also improve the durability of the system by moving the stripped end of the wires outside the squeezable part of the interface.

Compared to the study of Wang et al. [39] which sought to build strong connections between the conductive foam and electrodes, our method is more accessible. The fabrication of electrodes does not need access to a chemical laboratory, and the price of the electrode material is also much lower. The distribution of electrodes is also more flexible because they can be easily removed by cutting the knot and redistributing it in any shape. Compared to the methods that put a sheet of conductive fabric beneath or on top of a piece of conductive foam [15,27,28], from our experiment, the connection between the foam and the conductive fabric does not appear to be stable or sensitive (Table 1), although the contact area between them is large.

Compared to the past investigation of affordable electrode solutions [7] in commercially available conductive foam, our solution is more sensitive. We utilized contact resistance to increase its sensitivity to strain and load. We also conducted a deep investigation toward the stability and repeatability of each solution and provided a synthesis of each solution's performance.

In summary, the samples placed with conductive threads as electrodes, in which contact resistance between threads and foam was utilized, were significantly more sensitive to squeezes than other electrode solutions that were tested. On the other hand, samples with inserted electrodes are more stable and are capable of sensing deformation from different directions. Having taken our test result into account, deeper investigations were conducted into the dynamic resistance behaviors of the solutions using conductive threads.

No model was built in our study to predict strain or load according to $\Delta R / R_0$. Although the variance of the $\Delta R / R_0$ across measurements from the same sample is relatively smaller, it varies largely in measurements from different samples (Table A1). This is presumably caused by the subtle difference in the foam structures in different samples. Unlike foams manufactured in a lab using a 3D printer [12], in commercially available foam, the shape and size of the pores in a piece of foam vary and cannot be controlled. Hence, it is less suitable to allow the prediction of strain or load because of high variance. The question of how to process signals under manual compression is more practical and important for future practices.

## 7. Experiment 2: Dynamic Resistance Behaviors under Manual Compression

Two tests were conducted to observe the dynamic resistance changes in the foam sensors under manual compression during single compression (Test 1) and cyclic compression (Test 2). Test 1 was conducted to observe the resistance changes in the sample before, during, and after compression, providing sufficient non-compressed idle time. Test 2 was conducted to observe the resistance changes during multiple compressions with relatively short idle time. For both tests, three samples with two conductive threads inserted and three samples with two conductive threads placed on the surface of the foam were used.

In both tests, we opted for manual compression instead of machine-controlled compression. This choice aligns with the intended use of foam sensors by human users, as they might not be able to compress the sensor in a precisely controlled manner. Hence, it would be more meaningful to examine resistance characteristics under human compression.

### *7.1. Test 1: Single Impulse and Constant Compression*

The goal of this test was to investigate the dynamic resistance behavior under a single round of compression under two different compression patterns: "impulse" compression, which involved rapidly compressing and releasing the foam sensors like an impulse signal, and constant compression, as utilized in the previous experiment where the foam samples were compressed for a specific duration.

#### 7.1.1. Procedure

For the impulse compression pattern, the sample was first "in rest" for one minute, compressed manually using a plexiglass plate, and then released immediately. For the constant compression, the samples were first rested for one minute, manually pressed for approximately ten seconds with the plexiglass plate, and then released. In both compression patterns, after releasing the samples, the resistance data were measured for another minute to observe the resistance behaviors after compression.

In both experiments, the strain was 50% (15 mm deformation), controlled by the plexiglass placeholders (Figure 1c, 3). The aforementioned procedures for both compression patterns were repeated three times in all samples. The sampling rate was 100 Hz.

#### 7.1.2. Methods of Analysis

Since the absolute resistance values in rest for all samples were different, the relative resistance $R/R_0$ is used to compare the trend of resistance changes better. $R_0$ is the average resistance of the first 1500 data points when the sample was not deformed. The time evolution of $R/R_0$ for measurements in the same sample was plotted. For clarity and conciseness, measurements from one sample for each electrode solution will be presented in the next section. The data from the other samples can be found in Appendix D.

#### 7.1.3. Results and Discussion

**Response to Fast Manual Compression**. In both compression patterns, the resistance changed rapidly from the state of no compression to the state of being compressed and vice versa (Figures 6 and 7). More importantly, the samples with surface electrodes exhibited a drastic and rapid response (approximately 84% decrease in resistance) to quick compression. These properties enable both solutions to be used in real-time systems that need to respond quickly to fast user inputs. This characteristic was not shown in samples with surface electrodes. It is likely that, in such samples, contact resistance outweighs the resistance of the foam material.

**Characteristics in Each Stage of Compression**. The features that were discovered in the pilot study (Section 5.2) were also observed here in both patterns, such as the slight increase before compression, the surge of resistance after compression, slow recovery, and baseline drift. The drift in baseline varied in different measurements under the same deformation. Consistent with the findings from Experiment 1 (Section 6.4), the sample with surface electrodes exhibited high fluctuations when no compression was applied, whereas the signal became more stable during compression (Figures 6b and 7b). Relative resistance settled at similar values for the same sample and strain in both electrode solutions and both compression patterns. Additionally, from the results of impulse compression, it can be observed that both solutions were capable of capturing fast manual compression (Figure 6).

In summary, when the compression state changes, the resistance changes rapidly in both impulse and constant compression. This is manifested by steep downward and upward slopes. Noticeably, both solutions reacted fast to impulse compression, which was

not investigated and reported in the past study of electrode solutions based on commercially available foam [7].

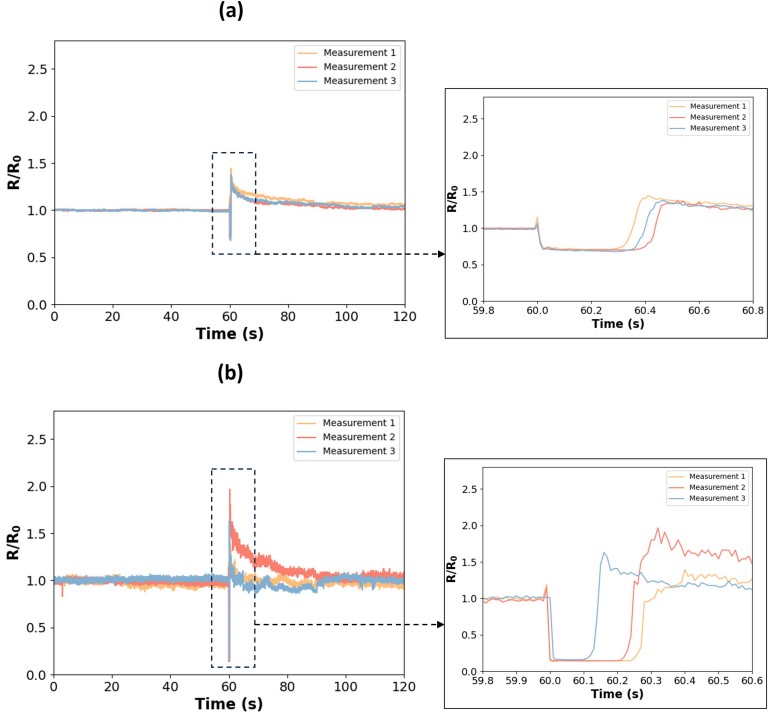

**Figure 6.** Measurements under impulse compression. Resistance dropped down when compression happened and surged when the sample was released. Electrode solutions: (**a**) inserted thread, 2 pieces; right: enlarged view of resistance behavior when the compression happened; (**b**) surface thread, 2 pieces; right: enlarged view of resistance behavior when the compression happened;

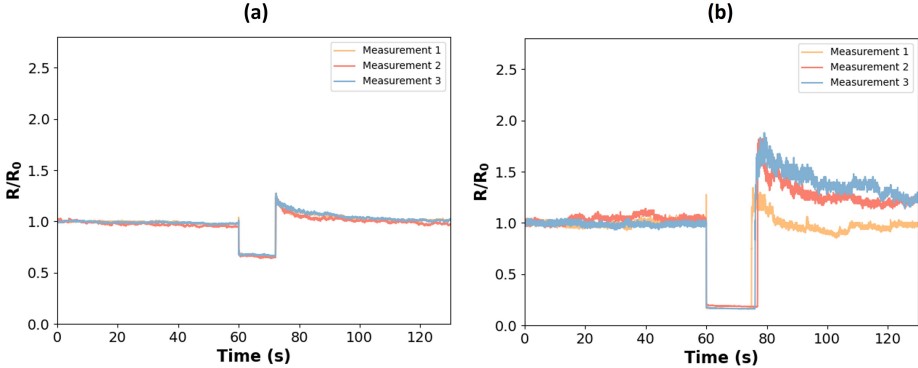

**Figure 7.** Measurements under constant compression. Resistance dropped down quickly when compression happened, remained low during compression, and increased when the sample was released. Electrode solutions: (**a**) inserted thread, 2 pieces; (**b**) surface thread, 2 pieces.

However, the problem of a drifting baseline can not be solved with the use of relative resistance only. In the next test, the objective was to observe the baseline drift and the consistency of resistance change under repeated compression to find a solution for the calibration of data under real-life application scenarios.

### 7.2. Test 2: Cyclic Compression

In this test, the goal was to investigate the dynamic resistance behavior of the samples when subjected to repeated compression. Our main focus was to observe the baseline drift and the dynamic characteristics of resistance change in the same sample during cyclic compression.

7.2.1. Procedure

The compression pattern used in this test was ten cycles of compression under the same strain, controlled by the test setup described in Section 3.2. One cycle consisted of approximately ten seconds of no compression followed by ten seconds of compression. A ten-second interval was used to control the release and removal of weights better. All ten cycles of compression were recorded continuously. The strain used in this test was 50% and a weight of 13.6 N was also used for compression. Similarly, the sampling rate was 100 Hz. The time to resistance figure was plotted for all samples separately.

To make the illustrations and descriptions more concise, one plot from each electrode solution will be displayed in the next section. The plots of other samples are included in Appendix D.

7.2.2. Results and Discussion

**Baseline Drift.** In measurements from the samples with inserted electrodes, the baseline significantly increased after the first cycle of compression and then fluctuated within a similar range in the subsequent cycles of compression (Figure 8a). The drastic increase in baseline after the first compression was not observed in the samples with surface electrodes (Figure 8b). Thus, this characteristic should be due to the characteristics of conductive foam material, but not the contact resistance.

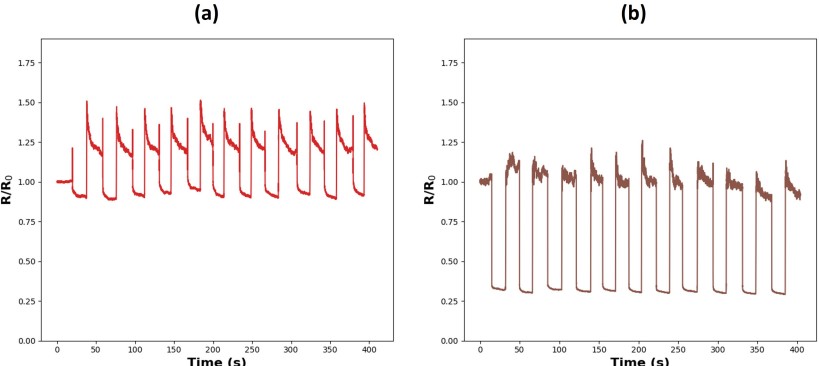

**Figure 8.** Cyclic compression. Procedure: ten seconds of rest, followed by ten seconds of compression, and release, repeated ten times. (**a**) Inserted thread, 2 pieces, sample 1; (**b**) surface thread, 2 pieces, sample 1.

**Consistency of Resistance Change.** The variance was relatively smaller in the samples with surface electrodes (Figure 8b), compared to the samples with inserted electrodes (Figure 8a). This is in line with the results in Experiment 1 (Table 1). The reason for the relatively more consistent reaction in the samples with surface electrodes might be the significant impact of contact resistance, which outweighed the resistance behavior of the conductive PU foam and is sensitive to load.

**Continuous Resistance Decrease**. In both electrode solutions, during the ten-second compression, there was a continuous decrease in relative resistance (Figure 8). This trend was particularly pronounced in the samples in which the conductive threads were inserted as electrodes. Similar to our findings in the pilot study, this behavior is likely caused by the creep behavior of foam. However, in samples with surface electrodes, the decreasing trend was less prominent in the plot, which might be caused by the more significant role that contact resistance plays in this electrode solution.

In summary, the resistance baseline of samples using inserted threads increased dramatically after the first compression, which implies that data calibration should be carried out after the first compression in such systems. Samples with surface electrodes had a more consistent reaction to the same strain and load.

## 8. Applications

To illustrate the potential of this sensing technology, two applications were built using conductive threads and conductive PU foam. All applications were based on Arduino Uno and Unity 3D.

### 8.1. Finger Interaction: Bubble Popping

This application was built to demonstrate the sensitivity of the solution with surface electrodes (Figure 9). Two electrodes were placed on the upper and bottom surfaces of a foam cube, respectively, and connected to the Arduino. The analog readings from Arduino are sent to Unity via serial communication. A gentle touch on the foam material generates a new soap bubble in this application. The pressure exerted during the squeeze directly influences the size of the bubble, with greater pressure leading to larger bubble sizes, up to a maximum limit. Upon releasing the foam, the soap bubble bursts. The calibration is conducted at the beginning. The users are expected to perform their hardest squeeze and hold for three seconds. Two baselines are established by sampling the average analog reading when no compression happens and the hardest compression is performed. Then, analog data are mapped to the interval [0,1] according to these baselines and control the size of the bubble. This technique can also be adapted to make other types of soft game controllers.

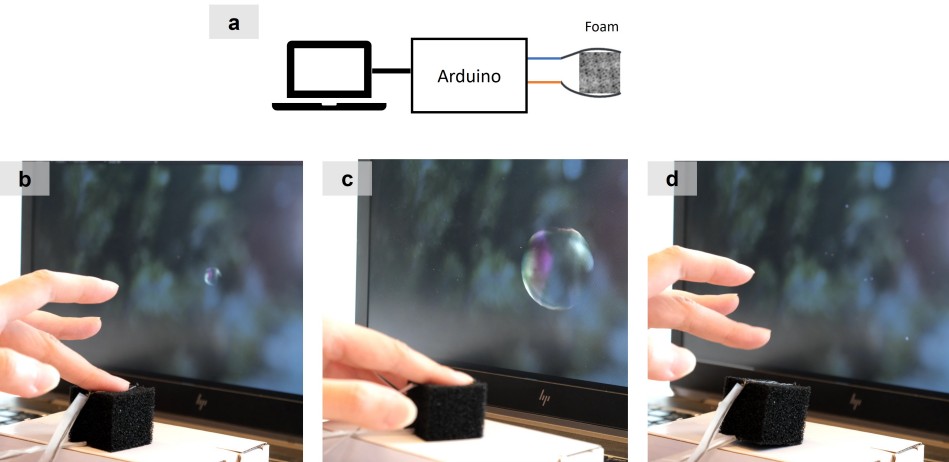

**Figure 9.** Application: Bubble popping. (**a**) System diagram. (**b**) A bubble is generated when a touch on the foam is detected, and (**c**) its size increases with the intensity of the squeeze. (**d**) The bubble pops when the foam sensor is released.

### 8.2. Hand Interaction: Interactive Stress Ball

This application was built to demonstrate its applicability for hand interaction and stress relief. This application was inspired by stress balls (Figure 10). In terms of hardware, the conductive foam was stacked into a spherical shape, with eight electrodes inserted into the foam ball and connected to the Arduino through an Unshielded Twisted Pair (UTP) cable, because squeezes need to be sensed from various directions. In terms of software, there are two circles on the screen: the left one shrinks and expands mimicking breathing, and the right one is controlled by the user. The harder the user squeezes, the smaller the circle on the right will be. When the pace of squeezes is similar to that of the circle on the left, two circles move toward each other, until they combine with each other. The calibration is performed at the start. The user is required to perform their hardest squeeze and hold for three seconds. Then, the average analog reading of the hardest squeeze and when the user releases the ball are calculated as the baselines to control the size of the circle on the screen.

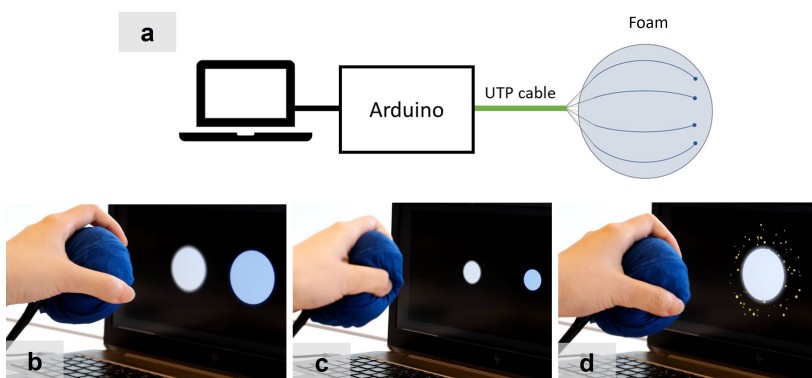

**Figure 10.** Application: Interactive Stress Ball. The size of the ball in the screen decreases with the hardness of squeezes. (**a**) System diagram. The other four electrodes are on the other side of the ball, They are hidden in the graph for clarity. (**b**) The ball is squeezed. (**c**) Two circles move towards each other when the squeeze is slow. (**d**) Two circles combine when they are close enough.

## 9. Discussion: Design Guidelines for Foam-Based Squeezable Interfaces

This chapter will discuss how our findings can guide researchers and DIY lovers to build soft interactive devices based on conductive foam.

### 9.1. Leveraging Contact Resistance Instead of Minimizing It

Our findings indicate that all the electrode solutions aimed at reducing contact resistance are much less sensitive than surface electrodes (Figure 5). This can be attributed to the fact that contact resistance is more sensitive to pressure [41]. By harnessing contact resistance, sensitivity can be significantly enhanced. Therefore, it is not always essential to minimize contact resistance when designing and developing interactive systems. For the same reason, when exploring electrode solutions, reducing resistance values does not always need to be the main focus, as smaller resistance values do not necessarily guarantee greater sensitivity compared to solutions with larger resistance values (Table 1 and Figure 5). Therefore, increasing the resistance change ratio is crucial for enhancing the sensitivity of an interactive system.

### 9.2. Choosing Electrode Solution Based on System Requirements

Surface electrodes are sensitive, which enables them to be employed in applications that require high sensitivity, such as light touch or subtle changes in squeezes. On the other hand, inserted electrodes can detect deformation from different directions with a few threads and have a more linear reaction to deformation.

### 9.3. Calibrating Data Based on Electrical Properties

Based on our findings regarding the dynamic properties of electrode solutions using conductive threads, we recommend employing two baselines to calibrate resistance data: the average resistance when there is no compression ($Baseline_0$), and the average resistance when the user performs their hardest squeeze ($Baseline_1$).

When inserted electrodes are used, the baseline significantly increases after the first compression (Section 7.1.3). Thus, it will be advisable to use the resistance after the first compression to calculate $Baseline_0$. By contrast, in samples using surface electrodes, this characteristic was not observed. Hence the calibration of $Baseline_0$ does not have to be restricted to before or after the first compression.

The resistance change for the same deformation is relatively consistent in both electrode solutions, indicating that the average resistance at maximum compression can be directly used as $Baseline_1$.

The resistance values can be mapped to the desired interval using these baselines. The resistance values that are bigger than $Baseline_0$ can be ignored since they are very likely the surges after compression (Section 7).

## 10. Conclusions

In this paper, a sensitive electrode solution has been proposed for detecting touches and squeezes by placing conductive threads on the surface of the foam. This solution utilizes the contact resistance between the foam and the electrode. Another electrode solution with conductive threads inserted into the foam was also investigated in both static and dynamic resistance behavior. It can sense squeezes from different directions. Based on our experiments, we observed the following characteristics of the dynamic resistance behavior of sensors based on commercially available conductive PU foam during compression: a slight increase at the onset of compression, a continuous decrease during compression, a surge at the end of compression, slow recovery, and drifted baseline, which can be observed in previous studies in other foam-based strain/pressure sensors as well.

Contact resistance, which was often seen as a factor that should be minimized, can considerably increase the sensitivity of such sensors. Its sensitivity to touches and squeezes makes the foam sensor sensitive to touch and squeeze interactions.

Drawing from the analysis of the static and dynamic properties of our solutions, we recommend embracing contact resistance, constructing sensors in alignment with system requirements, and calibrating signals based on the chosen electrode solution.

One limitation of this study is that only relatively small samples with a few threads were tested. Yet, it is in theory possible to build larger systems based on conductive threads, with big pieces of foam and multiple electrodes.

Another limitation is the lack of investigation under complex human manipulations, such as punching, hugging, or steering. Squeezes in real-life scenarios can be very energetic and unpredictable, thus this is an aspect that can be explored in the future.

Regarding application, recently, some studies have investigated squeezable devices for emotion regulation [18,19] and pain communication [21]. Furthermore, squeezing a stress ball is proven to be effective in regulating stress and anxiety [61,62]. Hence, there is an opportunity to leverage our approach to build squeezable stress relievers or squeezable communication devices when verbal communication is not possible (e.g., during dental treatment, meetings, and courses). Furthermore, several studies focusing on digital companions are also based on soft squeezable interfaces [20,63], and thus our study can also be applied to develop soft digital companions. As demonstrated in Section 8.1 and also from existing studies, such soft squeezable sensors can be used to develop game controllers [11,13,16]. Compared to commercialized force-sensitive resistors (FSRs), our solution does not introduce any stiffness and can be deformed in all directions. Moreover, by inserting conductive threads into the foam, the foam sensor can have a relatively linear reaction to deformation and can detect deformation in different directions (Appendix C). Such characteristics cannot be attained with FSRs or Velostat, since they are flat and thus cannot sense the amount of deformation.

In addition, using other soft materials to explore the comparison between capacitive sensing and resistive sensing could be an interesting direction for future research in the assessments of sensibility, stability, and repeatability. Such investigations have the potential to offer valuable insights regarding the advantages and disadvantages of resistive and capacitive sensing strategies when the same base material is used.

It is our hope that our work will provide insights to future researchers and makers who decide to use commercially available PU foam to sense touches or squeezes in their works.

**Author Contributions:** Conceptualization, L.G. and N.Z.; methodology, L.G. and N.Z.; software, N.Z.; validation, N.Z.; formal analysis, N.Z.; investigation, N.Z.; resources, L.G. and N.Z.; data curation, N.Z.; writing—original draft preparation, N.Z.; writing—review and editing, L.G., S.D. and N.Z.; visualization, N.Z.; supervision, S.D. and L.G.; project administration, L.G. and N.Z. All authors have read and agreed to the published version of the manuscript.

**Funding:** This research received no external funding.

**Institutional Review Board Statement:** Not applicable.

**Informed Consent Statement:** Not applicable.

**Data Availability Statement:** The data presented in this study are available on request from the corresponding authors. The data are not publicly available due to privacy and security concerns.

**Acknowledgments:** We would like to express our gratitude to everyone who helped with the experiments and the paper.

**Conflicts of Interest:** The authors declare no conflicts of interest.

## Abbreviations

The following abbreviations are used in this manuscript:

| | |
|---|---|
| PU | Polyurethane |
| HCI | Human–Computer Interaction |
| DIY | Do-It-Yourself |
| CV | Coefficient of Variation |
| UTP | Unshielded Twisted Pair |
| FSR | Force-Sensitive Resistor |

## Appendix A. Contact Resistance—Sample 2

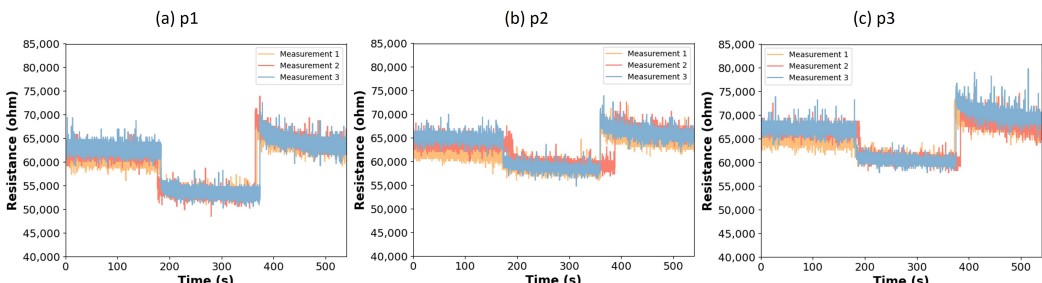

**Figure A1.** Resistance over time when a 5 kg weight was put on (**a**) p1—on the electrode, (**b**) p2—between an electrode and the center of the foam, and (**c**) p3—on the center of the foam.

## Appendix B. Table—Variance of Resistance Change Across Samples

Solution (e) and (f) exhibited relatively large variances in $\sigma(\Delta R_s/R_0)$ among measurements from all samples, especially in T2 (strain test). This can be attributed to different contact conditions between the foams and the threads due to the fabrication process, which leads to the variation in their contact resistance and its change range.

**Table A1.** $\sigma(\Delta R_s/R_0)$: the average SD of the resistance change ratios for measurements from all three samples under the same strain; $\sigma(\Delta R_l/R_0)$: the average SD of the resistance change ratios for measurements from all three samples under the same load.

| Electrode | $\sigma(\Delta R_s/R_0)$ (%) | $\sigma(\Delta R_l/R_0)$ (%) |
|---|---|---|
| a. Copper tape | 10.91 | 4.76 |
| b. Fabric tape | 6.21 | 4.61 |
| c. Thread (insert, 1 piece) | 5.34 | 3.79 |
| d. Thread (insert, 2 pieces) | 8.00 | 4.22 |
| e. Thread (surface, 1 piece) | 7.78 | 8.90 |
| f. Thread (surface, 2 pieces) | 6.64 | 6.60 |

**Appendix C. Experiments of Electrode Distance and Compression Direction**

This section will introduce the results of two experiments investigating electrode distance and compression direction. All samples used the solution of inserting two conductive threads as an electrode.

- Test 1 (T1): electrode distance
- Test 2 (T2): compression direction

*Appendix C.1. Procedure*

Three samples with electrode distances of 10 mm, 20 mm, and 30 mm were prepared for T1. Likewise, three samples with an electrode distance of 10 mm were used in T2, as the distances between each conductive thread were identical.

The testing procedure was one minute of resting and one minute of compressing. Regarding the orientation of sample placement, in T1, all of the samples were placed to enable compression on the threads (Figure A2a). Hence the distance between the two electrodes decreased when the sample was compressed. In T2, the samples were placed in a manner that the compression was not on the surface with electrodes (Figure A2b).

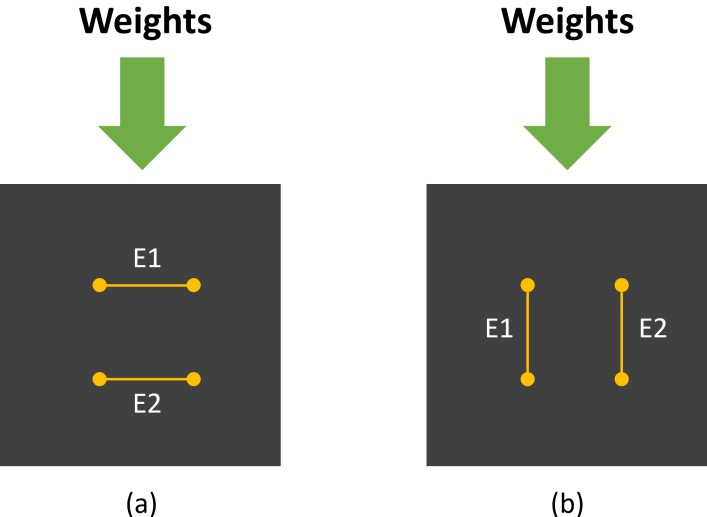

**Figure A2.** Orientation of sample placement in (**a**) T1: the distance between the electrodes was shortened when the sample was compressed; (**b**) T2: the distance between the electrodes was not shortened when the samples were compressed.

*Appendix C.2. Methods of Analysis*

Raw data were first segmented in Python to obtain the resistance data in 30 s before and after the onset of compression. Then, similar to the electrode experiment, the ratios of resistance change were calculated using the same method (see Formula (1)).

*Appendix C.3. Results*

All three electrode distances exhibited sensitivity within the same range of strain. It was observed that a larger electrode distance demonstrated slightly higher sensitivity in most of the cases (Figure A3a). Nevertheless, the average change ratio of the resistance of the samples using electrodes with a distance of 10 mm was close to that of the samples using electrodes with a distance of 30 mm in the highest strain (83%).

Figure A3b shows that there was not a significant difference in the ratio of resistance change between the two compression directions. This might be because, in the same strain, the contact areas between inserted threads and the foam were similar in both cases.

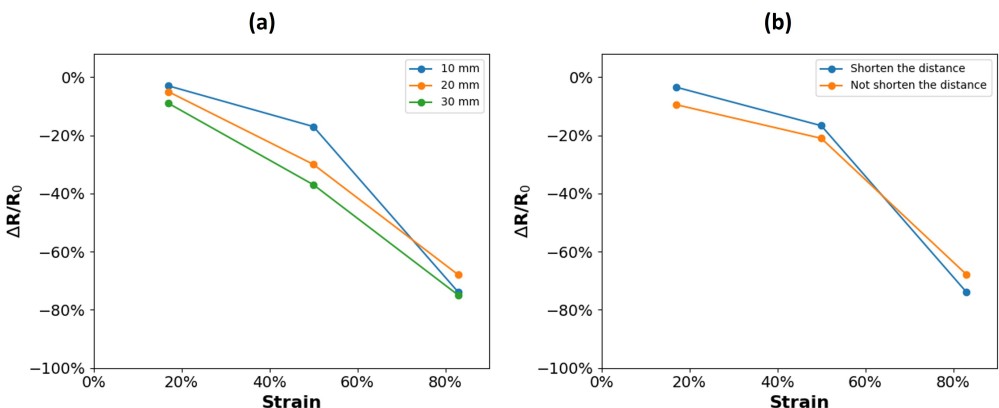

**Figure A3.** R: average resistance when the sample was being compressed; $R_0$: average resistance when the sample was not compressed. The first three plots show the influences on the resistance change ratio brought by (**a**) electrode distance and (**b**) compression direction.

**Appendix D. Supplementary Plots for Dynamic Resistance Behavior**

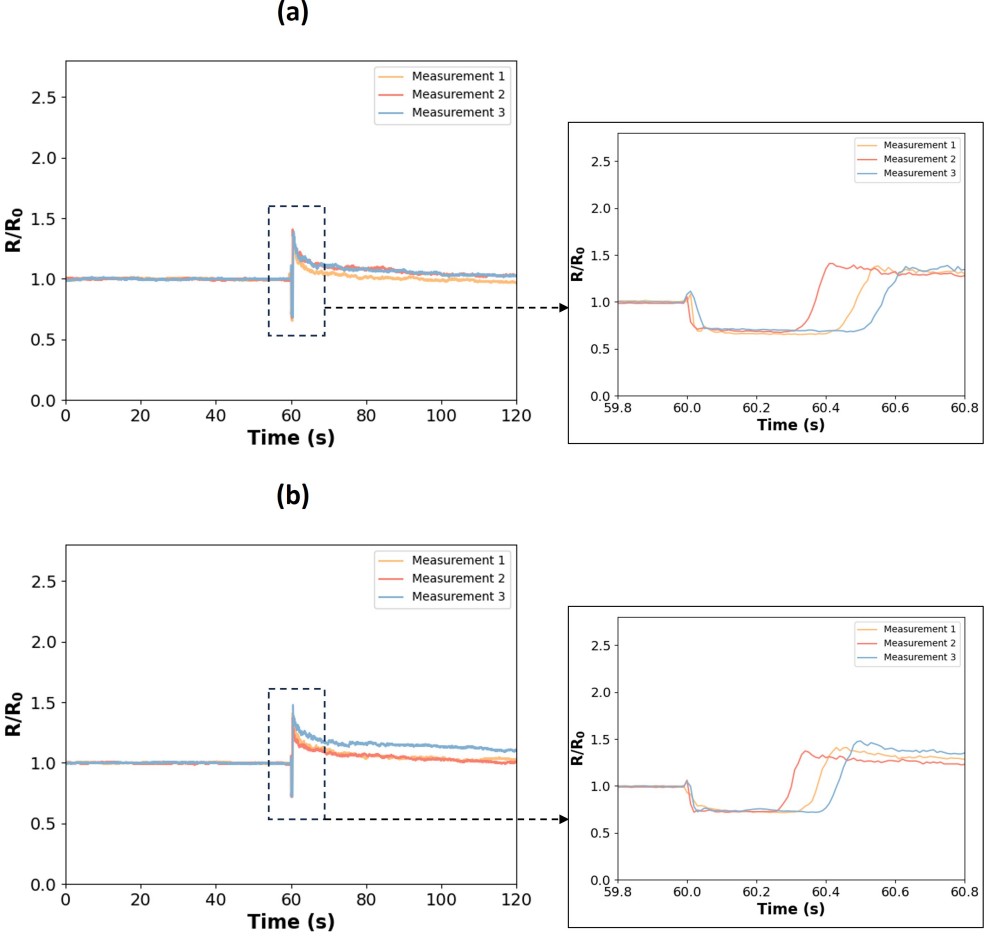

**Figure A4.** Impulse compression. Electrode solution: inserted thread, 2 pieces. (**a**) sample 1 (**b**) sample 3.

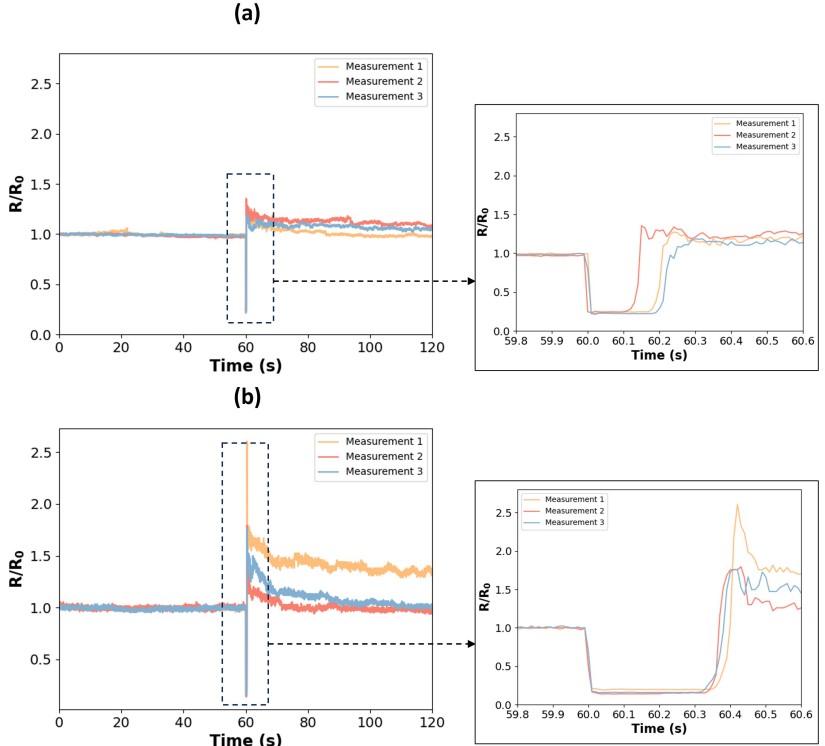

**Figure A5.** Impulse compression. Electrode solution: surface thread, 2 pieces. (**a**) sample 1 (**b**) sample 3.

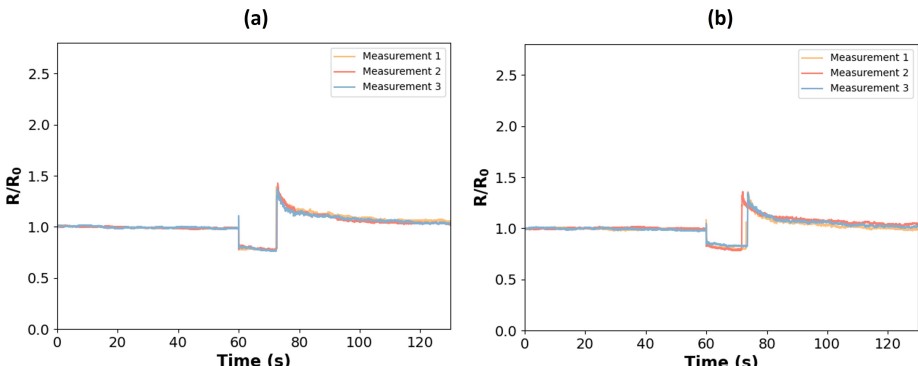

**Figure A6.** Constant compression. Electrode solution: inserted thread, 2 pieces. (**a**) sample 1 (**b**) sample 3.

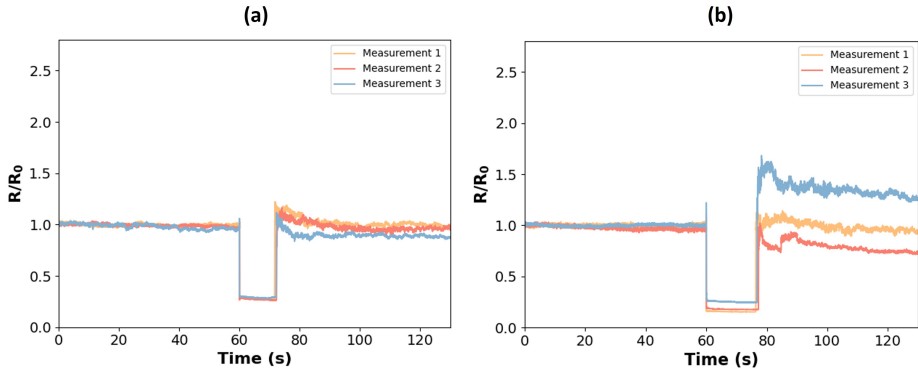

**Figure A7.** Constant compression. Electrode solution: surface thread, 2 pieces. (**a**) sample 1 (**b**) sample 3.

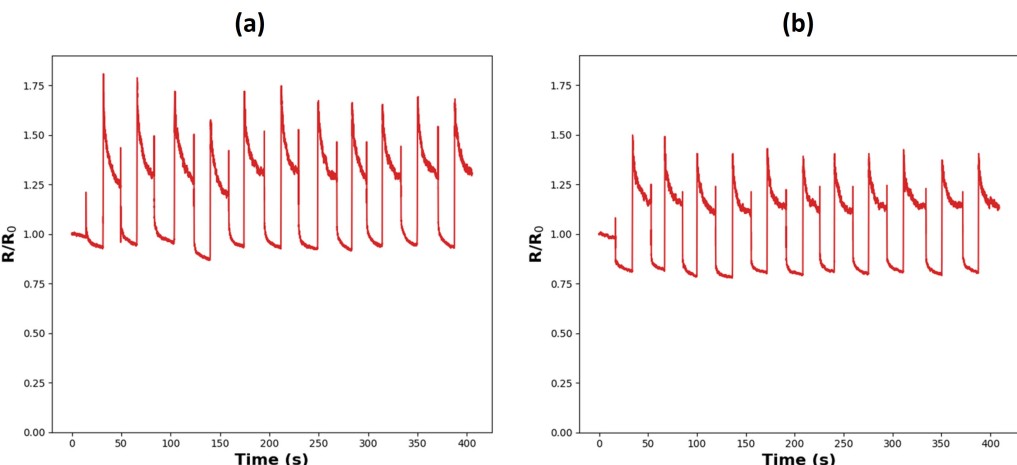

**Figure A8.** Cyclic compression. Electrode solution: inserted thread, 2 pieces. (**a**) sample 2 (**b**) sample 3.

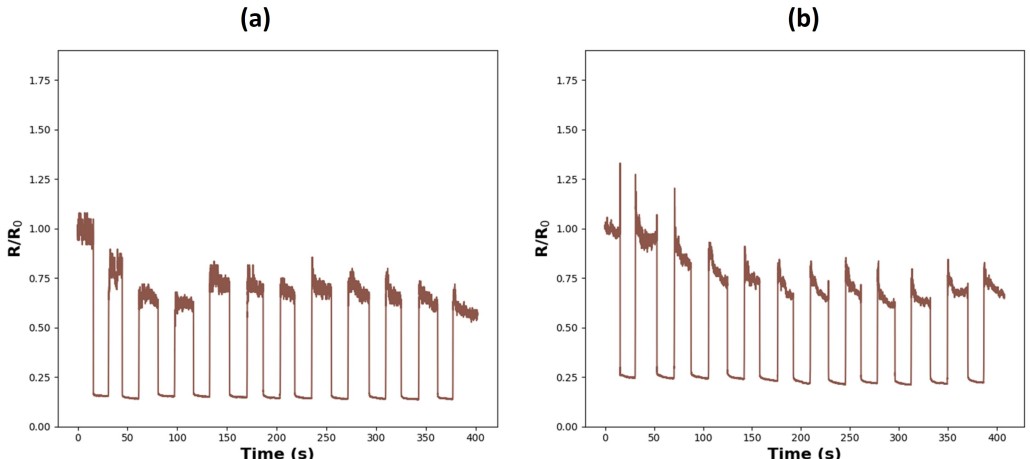

**Figure A9.** Cyclic compression. Electrode solution: surface thread, 2 pieces. (**a**) sample 2 (**b**) sample 3.

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
