# Peer review of "Contact Resistance Sensing for Touch and Squeeze Interactions"

_mti, doi:10.3390/mti8020014_

Round 1

Reviewer 1 Report

Comments and Suggestions for Authors

The article presents the study of a sensitive electrode solution for detecting touches and squeezes by placing conductive threads on the surface of foam. The study promotes the usage of the contact resistance between the foam and electrode.

The content is well presented using clear language and methodology.

The authors present basic measurements of resistance (variations) on several (limited) scenarios and configurations. More complex human manipulations are left for further investigation.

Is there any signal conditioning (filtering) performed before real processing (averaging)?

Some of the conclusions are introduced using the “suspect” and “may“ verbs, which doesn’t seem to be a strong rationale for the observed and measured behaviors.

The references seem a little bit outdated/old.  Only 14 out of 49 are from 2019 or newer. 

On line 253, is 0.25 mm correct? Isn’t it too small?

On line 573, “hardness” must be replaced by “hardest”.

Overall, the article presents a rather simple (light conclusions)  but still solid (fair methodology) study which may prove itself valuable for further investigations.

Author Response

Dear reviewer,

Thank you very much for the time you dedicated to reviewing our paper. Please find our responses in the attachment.

We hope that these revisions address your concerns, and we are hopeful that the manuscript is now more polished for publication.

Kind regards,

The authors

Reviewer 2 Report

Comments and Suggestions for Authors

Nowadays DIY technologies have advanced far enough concerning situation that has been observed in 80-ns of the previous century or even at the beginning of 2000-s.

3D printers, microcontrollers and easy access for acquisition of advanced materials give a freedom for scientific creativity and implementation of crazy projects. The most difficulties and limitations which are on a way of researchers and enthusiasts are the budget, time frames to collect the necessary components due to logistical force-majeure circumstances and of course personal skills and experience. The most difficulties and limitations which are on a way of researchers and enthusiasts are their budget, time frames to collect the necessary components due to logistical force-majeure circumstances and of course personal skills and experience. Yet, thanks to youtube and multiple thematic forums to get reasonable advices in any field of technology is not a problem. Therefore, the fear of the necessity of “access to a chemistry laboratory” which emphasized by the authors several times in Introduction section cannot be the serious reason of avoiding at the development of reliable sensors.

On the other side, the reader having an interest in development “Contact Sensing for Touch and Squeeze Interactions” is expecting to get much more different information not only related to a couple of case studies in experimenting with easy affordable materials and fabrication technologies, measurement and data processing with a DC conductivity of PU porous foam. It is expected to get advices how to improve/stabilize the results of measurement due to the contact between lead-off wires and sensitive to pressure foam material. For instance, the authors could recommend use DIY conductive glue, or gel, or specific wires configuration at the contact place.

The authors also shortly mentioned in the Related work section a capacitive sensing that supposed avoiding the DC contact problems. But why the authors did not presented comparative characteristics (pros and cons) of two quite similar in their simplicity (Fig. 1b) for the measurement DC conductivity and capacitance (see reference below known from 2014 being in Arduino IDE lib. search for "capacitor") schematic solutions based on Arduino microcontroller.

Capacitance measurement with the Arduino Uno. Available at (accessed on 30.12.2023) https://codewriteuk.wordpress.com/2014/01/21/cap-meter-with-arduino-uno/

A comparative sensibility, stability, and repeatability of two DIY technologies would be much more interesting to the readers, than just most information how to avoid doing in a way as tested/presented by authors.

Even a comparison of the stress ball like software interfaces implemented with two types of sensors would be much more interesting for the readers.

Author Response

Dear reviewer,

We would like to express our gratitude for your time and efforts in reviewing our paper. We appreciate the insights you have shared.

We have carefully considered your comments and suggestions, and we have addressed some of the key points you raised in our manuscript.  Please find our responses to your comments in the attachment. 

Thank you once again for your time and valuable input.

Kind regards,

The authors

Reviewer 3 Report

Comments and Suggestions for Authors

This study proposes an electrode solution to detect touches and squeezes with conductive threads and PU foam. Especially, the idea of utilizing contact resistance rather than minimizing it is very interesting and unique. The authors showed static and dynamic resistance characteristics and their explanations in detail. As the authors showed in the applications, the proposed solution is expected to provide sensing technology in various fields.

There are some comments and questions as follows.

(1) Figure 1(a) and 1 (c) are interchanged in the main text.

(2) In the pilot experiments, the details of the materials such as model and make are not clear.

(3) In the pilot experiment for contact resistance, the height of the plexiglass pile is written, but its weight is missing, which is thought to be important.

(4) The captions and mentions in the text of Figure 3(a) and 1 (b) are interchanged.

(5) A slight increase at the onset of compression is not noticeable in Figure 3(c). It is quite noticeable in Figure A1.

(6) In experiment 1: electrode solution, the way to control the strain is not clear. How did the authors control the various strain values?

Sincerely,

The reviewer.

Author Response

(The authors gave the same response as above.)

Reviewer 4 Report

Comments and Suggestions for Authors

The paper proposed a sensitive electrode solution to detect touches and squeezes by placing conductive wires on the surface of the foam, which utilized the contact resistance between the foam and the electrode.

The topic is interesting, but it is an academic approach without a direct identification of applicability in a real application scenario. The authors should present some applicability use cases, as well as identify what the added value is in relation to the commercialized solutions.

Regarding Figure 1, some additional information is necessary. In subfigure a) the wires need to be identified and labeled, in particular, 5V, GND and signal. In subfigure b) a voltage divider configuration is shown, and it is important to identify that the foam sensor is resistive and explain why this type of configuration is necessary with a resistance of 10KOHM. subfigure c) identifiers 1, 2 and 3 need to be further explained in particular what is the relationship with the foam resistance or the voltage output that is injected into the Arduino.

Avoid writing in the first person – “We”.

The paper is very long, with an exaggerated number of chapters. 11 chapters for this type of paper is too much. On the other hand, future work should be integrated into the concluding chapter or after, and not before, as presented in the paper.

Author Response

(The authors gave the same response as above.)

Round 2

Reviewer 2 Report

Comments and Suggestions for Authors

proof-reading would be highly recommended, e.g.:

Line 213 Similarly, the surge in resistance after compression can also be caused by the same reason.

= The same/similar factor can cause the surge in resistance after compression.

Comments on the Quality of English Language

No

Author Response

Dear reviewer,

We appreciate your thorough review and suggestion regarding the clarity and readability of our paper. 

Regarding your suggestion on line 213, we have carefully considered your recommendation. We have not only revised this sentence (on page 6, lines 210-211)  but also involved a former colleague to enhance the readability of the paper. We have highlighted all revised parts in the new manuscript.

We believe this modification improves the overall clarity of the expression. We hope that these changes address the concerns you raised.

Thank you once again for your time and dedication to reviewing our paper. 

Best regards,

Nianmei, Steven and Luc